# Immunoglobulin-Storing Histiocytosis: A Case Based Systemic Review

**DOI:** 10.3390/jcm10091834

**Published:** 2021-04-23

**Authors:** Hanne Wiese-Hansen, Friedemann Leh, Anette Lodvir Hemsing, Håkon Reikvam

**Affiliations:** 1Institute of Clinical Science, Faculty of Medicine University of Bergen, N-5021 Bergen, Norway; hanne@datagrafikk.no (H.W.-H.); Anette.Hemsing@uib.no (A.L.H.); 2Department of Pathology, Haukeland University Hospital, N-5021 Bergen, Norway; friedemann.leh@helse-bergen.no; 3Department of Medicine, Haukeland University Hospital, N-5021 Bergen, Norway

**Keywords:** storing histiocytosis, crystal-storing histiocytosis, immunoglobulin, MGUS 5, B-cell neoplasia

## Abstract

Crystal-storing histiocytosis (CSH) is a rare event in disorders associated with monoclonal gammopathy and is mostly associated with the accumulation of immunoglobulins (Igs) in the cytoplasm of histiocytes. In this article, we present a case of a 75-year-old female with IgG kappa monoclonal gammopathy of undetermined significance (MGUS) and signs of a non-crystallized version of immunoglobulin-storing histiocytosis (IgSH) in a vertebra corpus. Furthermore, we performed a literature review based on all cases of storing histiocytosis identified by literature search between 1987 and 2020 and identified 140 cases in total. The median age at diagnosis was 60 years (range 18–91), with an equal sex distribution (51% men). The majority of the patients had an underlying neoplastic B-cell disorder, most often multiple myeloma (MM), MGUS, or lymphoplasmacytic lymphoma (LPL). The main affected organ systems or tissue sites were bone (*n* = 52), followed by head and neck (*n* = 31), kidney (*n* = 23), lung (*n* = 20), and gastrointestinal (GI)-tract (*n* = 18). IgG was the main immunoglobulin class involved, and most cases were associated with kappa light chain expression. We conclude that IgSH is a rare disease entity but should be considered with unusual findings in several organ systems associated with monoclonal gammopathy, especially with kappa light chain expression.

## 1. Introduction

Crystal-storing histiocytosis (CSH) is a rare disorder characterized by the accumulation of crystallized deposits in the cytoplasm of histiocytes. The entity has been associated with underlying lymphoproliferative or plasma cell disorders, such as monoclonal gammopathy of undetermined significance (MGUS), multiple myeloma (MM), or lymphoplasmacytic lymphoma (LPL) [1,2,3]. The histiocytes in CSH contain crystallized material, although there have been reported cases of immunoglobulin-storing histiocytosis (IgSH) without a crystallized pattern of the deposited immunoglobulins [4,5]. IgSH can present in both a localized and a generalized form and may include a wide range of tissue sites and organs [1,2]. Herein, we present a patient with increasing back pain after a minor injury. She was diagnosed with a localized form of non-crystallized IgSH. We discuss the clinical findings, diagnostic workup, and therapeutic options. Furthermore, we performed a systematic review of the literature regarding IgSH and summarized the findings in this rare disease entity.

## 2. Case Report

A 75-year-old woman presented with acute pain in the lower back after a trivial incident. Her past medical history included hypertension, hypercholesterolemia, type 2 diabetes mellitus, hypothyroidism, migraine, and arrhythmia. Initial radiological examinations demonstrated no fracture. However, over the next few weeks, her back pain increased. By magnetic resonance imaging (MRI), she was diagnosed with a compression fracture in L1, although with no spinal stenosis (Figure 1). The fracture was initially managed conservatively; however, she had persistent opioid-dependent lower back pain and was admitted to the orthopedic department for further clinical and laboratory diagnostic workup to exclude a pathologic etiology for her fracture.

When blood tests revealed anemia, elevated sedimentation rate (SR), and a monoclonal (M)-protein by serum electrophoresis (Table 1), she was transferred to the hematology unit, as MM or other plasma cell dyscrasia was suspected. A bone marrow aspirate was performed and demonstrated a slightly hypocellular bone marrow with megakaryocytes present. There was normal maturation in the erythrocytopoiesis and the granulocytopoiesis, without expansion of lymphoid cells. The plasma cells accounted for 4% of nucleated marrow cells. Furthermore, whole-body low dose computed tomography (CT) for assessment of osteolytic lesion as part of MM was performed without detecting other lesions. Hence, the patient did not fulfill the diagnostic criteria for MM. However, based on the findings of monoclonal gammopathy by serum electrophoresis and the absence of findings supporting MM, she was diagnosed with MGUS. For further diagnostic workup, a CT-guided biopsy of the lesion in L1 was performed.

The bone biopsy showed only sparse monoclonal plasma cells, but vast amounts of deposits intracellularly stored in histiocytes, inconsistent with amyloid (Figure 2). In immunohistochemical staining for light chains, the deposits were somewhat stronger positive for kappa than for lambda. In additional stains, IgG was colocalized with kappa and CD68, making it probable that the deposits consisted of whole immunoglobulins. By electron microscopic examination, there was abundant bright, predominantly amorphous material, but also some tubular structures with a diameter between 25–30 nm. No relation to cell organs was seen. Additionally, small amounts of electron-dense amorphous material were found. Attempts with mass spectrometry gave no conclusive results.

After clinical, radiological, histopathological, and biochemical examination, it was concluded that the patient had MGUS with secondary IgSH, without any other underlying malignant disease. The patient had no sure sign of CRAB-criteria associated with MM: hypercalcemia, renal failure, anemia with Hb < 10 g/dL, or osteolytic bone disease. The patient, on the other hand, had persistent pain and difficulty moving, and she was, therefore, accepted for surgical intervention with vertebroplasty. She was operated on with the insertion of pedicle screws in vertebra corpora Th12 and L2, followed by repositioning of the fracture in L1 and the insertion of cement in the corpora with access via the pedicles bilaterally. The peri- and postoperative courses were without complications. The patient had striking relief of pain after surgery; she was no longer dependent on analgesics and could resume normal daily life activities. The current follow-up by the back surgeon and hematologist for her MGUS does not show signs of disease progression.

## 3. Literature Review

### 3.1. Methods and Classification

We performed a systematic PubMed search with the term “Storing Histiocytosis” in the title and identified 107 relevant published articles from 1987 to July 2020. The term “Storing Histiocytosis” was used to include both CSH and IgSH. Only cases with sufficient information written in English were included, and six articles were thereby rejected. From 101 articles, we identified 140 cases of assumed IgSH (Table 2). Other data collected were the type of material deposited within histiocytes, age, gender, involved sites, type of immunoglobulin if available, and the presence or not of an underlying lymphoproliferative disease or plasma cell dyscrasia (Table 2).

### 3.2. Classification and Etiology

The median age at diagnosis was 60.5 years for both sexes (age-range women 18–91, age-range men 32–86), and there was a nearly equal sex distribution with 71 men (51%) and 69 women (49%) (Table 3). The age and gender distribution are presented in Figure 3. Cases were divided into two subgroups: localized and generalized, as proposed by Dogan et al. in 2012 [55]. Cases with only one involved site were characterized as localized (77%), while cases with two or more involved sites were classified as generalized (23%) (Table 3).

### 3.3. Organ Affection

IgSH can be found in a wide range of organs/tissues. The most frequently involved organ systems and tissue sites included bone, head and neck, kidney, lung, gastrointestinal mucosa, and lymph node. The organ and tissue site involvement in the identified cases is presented in Figure 4.

### 3.4. Immunoglobulin Classes and Immunoglobulin Restriction

Immunoglobulin classes were collected in this literature review and are presented in Figure 5. The information regarding the specific immunoglobulin and light chain present in the serum was not always available or specified. The most common immunoglobulin found by serum protein electrophoresis and immunofixation was IgG identified in 39 cases, while IgM and IgA were identified in 20 and 12 cases, respectively. The presence of light chains only was identified in nine cases (Figure 5). Hence, in most cases, the monoclonal spike was made of both heavy and light chains.

The type of light chain, i.e., either kappa or lambda light chains, is presented in Figure 6. Kappa light chain was the most common light chain associated with IgSH (65 cases), either as part of complete Ig or as free light chain alone (Figure 6). Lambda light chain was only found in 10 cases.

### 3.5. Etiology of Material Deposited

To include both non-crystallized and crystallized material within histiocytes, the term Storing Histiocytosis was used in the literature search. Three cases, including our present case, were described as “non-crystallizing” by histopathological review. On immunohistochemical analysis, the crystals are normally monoclonal, with eight exceptions in this review (Table 2). In many articles, information regarding the etiology of the crystals was insufficient and thereby classified as not available (NA). 6% had negative immunohistochemical staining for immunoglobulin, but many had an associated neoplastic B-cell disorder and had positive staining of plasma cells and lymphocytes in the surrounding area of the histiocytes and were described as an immunoglobulin-type of CSH.

## 4. Discussion

CSH is a very rare disease ethnicity. Therefore, very few studies related to this condition exist, and the knowledge in the field is largely based on case reporters or smaller patient series. Guidelines for diagnosis and treatment are almost non-existent. Increased knowledge and understanding about this condition are therefore welcome to the medical community in order to increase insight into this condition.

In the present paper, we describe a case of non-crystallized IgSH with, as far as we know, only two other cases previously presented in the literature [4,5]. IgSH has been linked to CSH, which is a rare entity characterized by the accumulation of crystallized material within histiocytes [1,55]. CSH can be classified according to crystal deposited, or according to etiology and associated disease, often underlying B-cell neoplasia or rarely allergic-autoimmune, drugs, metabolic or inflammatory-reactive [55] (Table 4). Immunoglobulins are the most common material deposited in CSH, with kappa being the dominant light chain [11], as also found in our study (Figure 7). Other rare types of crystallized material have been reported, including clofazimine crystals, Charcot–Leyden crystals, cystinosis, and crystallization due to exposure to silica [97,98,99,100]. Due to the finding of non-crystallized IgSH, we propose to alter the classification as shown in Table 5.

The term “Storing Histiocytosis” was used to include non-crystallized versions of IgSH in this literature review. A total of 140 cases were identified through a PubMed Search dating from 1987 to July 2020. The crystallized material was made up of clofazimine crystals, Charcot–Leyden crystals or amyloid material in six CSH cases found in our literature search. These cases were thereby not included in the review due to their non-immunoglobulin material. We included 5 cases in our literature search that were “triple-negative,” i.e., the articles did not specify if a lymphoproliferative–plasma cell disorder was present, the type of immunoglobulin in serum, or the type of immunoglobulin within histiocytes. These were included since we considered it likely that the cases were immunoglobulin-related [27,35,42,50,60]. Woeherer et al. presented a 55-year-old male patient with a CNS tumor, which is spoken about as immunoglobulin-related CSH, although the material is not specifically characterized as immunoglobulin [27]. Vaid et al. presented a patient with a gastrointestinal CSH, where a lymphoproliferative disease was suspected and is thereby included [35]. Another case was presented, where there was an association with lymphoplasmacytic infiltrate, though not sufficient to diagnose an LP–PCD [42]. Kawano et al. described their material as crystallized immunoglobulin without specifying the immunoglobulin restriction within the histiocytes [50]. Kamimsky et al. presented a case of CNS CSH, in which the etiology behind the crystallized material was unknown. This case was included due to the possibility of it being immunoglobulin-related; while further examination was not performed, other possible materials were thought of and ruled out [60]. Crystallized immunoglobulin was found in 90 cases, while non-crystallized immunoglobulin was found in three cases [4,5]. Of the cases reviewed, 87% had an underlying neoplastic B-cell disorder, with MM being the most frequent (43 cases, 31%). IgG was the main immunoglobulin class involved, and most of the cases were associated with kappa light chain expression (Figure 6).

IgSH can be found in a wide range of organs/tissue sites and is divided into generalized and localized forms based on the number of affected organs/tissue sites. We found 32 cases (23%) of generalized disease and 108 cases (77%) of localized disease. Bone involvement was the most frequently involved organ/tissue site in our review. Dogan et al. performed an extensive review of CSH in 2012 and found that head and neck were the most frequently affected sites in the localized form and bone marrow in the generalized form [55]. The mentioned review classified 58% of the cases as localized disease, which is lower than in the present review [55]. This difference may be due to increased awareness of this entity in recent years, leading to diagnosis at an earlier time in the disease development.

CSH has been associated with underlying B-cell neoplasia in as many as 90% of the cases in earlier reviews and 87% of the IgSH cases in this review [55] (Table 2). Earlier detection of IgSH in recent years may also explain the lower percentage of LP–PCD in this literature review. Lack of complete staging in newer case reports or loss of patients to follow-up may also explain the difference in frequency. The B-cell neoplasia’s associated include the main secretory B-cell malignancies; MGUS, MM, and lymphoplasmacytic lymphoma (LPL) [1,2,3,101].

Clofazimine, a drug used to treat leprosy, has been linked to the development of CSH in two cases [102,103]. CSH has also been reported in cases of benign plasma cell proliferation such as hypergammaglobulinemia and plasma cell granuloma, which may indicate that CSH is a result of high values of abnormal immunoglobulins rather than being a result of lymphoproliferative disease [85,88].

A patient, reported by Uthamalingam et al., presented with similar symptoms as our patient with worsening pain around the hip after a trivial fall some months earlier [22]. An MRI showed multiple compression fractures of the lumbar vertebra and a trochanteric fracture. The patient also suffered from anemia. A plasma cell dyscrasia was suspected, and serum electrophoresis demonstrated an M-component, while immunofixation identified them to be IgG kappa. A bone marrow aspirate showed both mature and immature plasma cells and also a few large histiocytes with long cytoplasmic crystals. Extensive workup led to the diagnosis of MM. Histological examination of the excised femoral head showed abundant histiocytes with intracytoplasmic crystallized material leading to the diagnosis of CSH. Since fractures in the axial skeleton are uncommon in plasma cell myeloma, the fracture of the femoral head was suspected to be caused by CSH [22]. This case has many similarities to the case presented, making common pathogenesis probable.

Regarding cases of IgSH without crystallization, very few cases have been reported. The first known case of a non-crystallized form of IgSH in the lungs was described by Chantranuwat et al. in 2007. The patient presented with dyspnea, fever, and bilateral patchy lung infiltrations whilst on chemotherapy for MM. A lung biopsy was performed demonstrating intra-alveolar accumulation of macrophages with round eosinophilic globules in the cytoplasm. These globules stained positively for kappa light chain expression. Electron microscopic examination showed no linear parallel configuration of the immunoglobulins as is seen in CSH and had several comparisons to our reported case [4]. The second case of non-crystallized IgSH was presented by Kurabayashi et al. in 2010 in a 53-year-old woman with Sjogren’s disease and MALT thymic lymphoma. When examining the biopsy of the thymoma, the accumulation of eosinophilic histiocytes was discovered, which were positive for IgG kappa [5]. All three cases showed an accumulation of kappa light chains within the histiocytes, which increases the probability of its association with CSH and common pathogenesis. Electron microscopic examination of our case showed abundant bright material within histiocytes but without a crystallized pattern with rhomboidal, needle-shaped, or parallel arrays of crystals as documented in the literature for CSH; thereby, it is an important examination to distinguish the crystallized from the non-crystallized immunoglobulins [4,13,17]. Chantranuwat, who presented the first case of IgSH, proposed to use the classification of ISH widely with subclassification of either crystallized or non-crystallized immunoglobulins [4]. We propose to use the term IgSH instead of ISH due to the widely known abbreviation of ISH to describe in situ hybridization. The histiocytes in CSH are usually strongly positive for CD68 and negative for desmin, myoglobin, muscle-specific actin, CD1a, and S-100 protein. The crystals are stained blue with phosphotungstic acid hematoxylin and positive in periodic acid-Schiff stains [18,55]. On immunohistochemical analysis, the crystals may be monoclonal, polyclonal, or not stain at all. Suboptimal tissue fixation antigen-masking resulting from the crystalline structure of the protein or altered molecular formation may lead to failure to stain [18,55]. Given the possibility of faint or negative staining on immunohistochemical analysis, the information in the clinical history regarding the presence of a serum monoclonal protein may be crucial in the identification of the deposited material as immunoglobulins [81]. The material within the reactive histiocytes is most often kappa light chains. When M-proteins are not found by regular serum immunofixation, assays for serum free-light chain identification can be used.

The pathophysiology behind the crystal formation remains unclear, with theories regarding overproduction or failure to degrade immunoglobulins intracellularly. Several studies show no or low paraprotein levels in the blood of patients diagnosed with CSH, making the theory of overproduction less likely [18]. It has also been hypothesized that DNA mutations in the sequence for immunoglobulins may result in resistance towards lysosomal degradation by macrophages [17].

Most often, the patients with CSH/IgSH present with an asymptomatic mass or swelling, although other symptoms may occur [55]. Some patients also present with symptoms such as fever or other general symptoms, revealing CSH as a more systemic inflammatory syndrome [13,26,57]. Noteworthy, this could also lead to elevated inflammatory markers, such as CRP and ferritin, and inflammatory anemia [104]. Inflammation of the serous tissues, serositis, can also occur as part of the disease. Lesesve et al. presented a woman with CSH found in ascites, while Galed-Placed et al. presented a case of CSH and MM in a pleural effusion [58,76]. In our case, the patient presented with increasing back pain, and the diagnosis was made because of her uncommon progression of a compression fracture in the L1-vertebra.

Treatment and prognosis of patients with CSH vary according to the associated underlying disease. Given the rarity of CSH, clinical studies regarding optimal treatment approaches are lacking, including the specific response of CSH following chemotherapy or simple excision. Therefore, individual therapeutic approaches based on the patient’s symptoms and disease burden are used for practical purposes. If the patient’s underlying B-cell malignancy requires treatment, such as symptomatic MM or LPL, most physicians will probably treat the disease according to the appropriate treatment algorithm. As a consequence of CSH’s association with an underlying disease, research regarding therapeutic options is limited as the symptoms following CSH decrease when the underlying cause is successfully treated [55]. However, CSH can show persistence in follow-up biopsies after chemotherapy and stem cell transplantation [85]. Local treatment with surgery or other interventions, as in our present patient, should hence be considered special for patients with localized CSH without treatment requiring systemic disease. The number of foci of CSH probably also has a prognostic impact, and patients with generalized CSH tend to have the worst prognosis than those localized CSH [3]. The symptoms of CSH can lead patients with underlying B-cell malignancies to earlier diagnostic workup CSH [3] and hence lead to diagnosis at an earlier stage of disease than would otherwise occur, leading to a better prognosis. The discovery of CSH should therefore lead to investigations to dismiss diagnoses such as MM, MGUS, LPL, or other B-cell malignancies, as many of the cases are associated with a lymphoid or plasma cell neoplasm [7,12,22]. Though the impact CSH has on the prognosis remains unclear, it seems that patients with MM and CSH have reported survival 5–15 years longer than MM patients without CSH [85,105]. Classification of CSH and disease etiology (Table 3 and Table 4) may have the potential to more uniform diagnostic algorithms and treatment decisions; therefore, clinical cases and studies are easier to compare.

Gaucher’s disease, a histiocytic storage disease, needs to be assessed as a differential diagnosis, especially when histiocytes with crystallized material are found in the bone marrow [75,85]. Gaucher’s disease is a metabolic storage disease that resembles the needle-like accumulation of immunoglobulins in the cytoplasm of histiocytes in CSH, with fine cytoplasmic striations caused by the accumulation of glucocerebroside. As opposed to the histiocytes in CSH, Gaucher cells normally stain brightly positive for iron, but when in doubt and for exact diagnosis, assays for β-glucocerebrosidase should be performed to properly differentiate [106]. There are several other diseases with a histiocytic or histiocyte-like infiltrate that need to be excluded [55].

A few cases of CSH have been reported due to clofazimine treatment in patients with leprosy, where a good clinical history and examination may be of more value in the diagnostic process than a biopsy. This finding led to the proposition to use the term clofazimine-induced CSH by Sukpanichnant et al. [102,103]. CSH associated with massive deposits of Charcot–Leyden crystals have been reported a few times, one causing colonic polyps in a 78-year-old woman and another as systemic mastocytosis in the bone marrow of a 91-year-old woman [97,98]. CSH in association with cystinosis has been described once by Gebrail et al. in a 23-year-old man with hereditary cystinosis [99]. A bone marrow biopsy was performed because of an abnormal blood count, which showed clusters of hexagonal, tubular, and rectangular cysteine crystals within macrophages. Weiss et al. described in 1978 the development of a fibrous histiocytoma in seven patients, who had been exposed to silica, most of them after injections to repair hernias [100]. Macrophages with intra- and extracellular crystals, identified as silica by x-ray diffraction, were seen in all lesions. Cho et al. presented a case of CSH, associated with MM, as a possible result of prolonged treatment with carbamazepine, an antiepileptic drug, for the management of peripheral neuropathy [19]. Cases of plasma cell neoplasm have been reported with carbamazepine exposure, but the mentioned case report addresses the first possible related CSH [19]. When serum protein electrophoresis is negative, and immunohistochemistry fails to stain the intracytoplasmic crystals, diagnostic challenges arise. Ko et al. described the diagnostic difficulties in separating immunoglobulin crystals from mycobacteria [53]. Kaminsky et al. presented a case with unknown crystal etiology and origin, debating whether their case of CSH may be due to immunoglobulin deposits or due to Pentasa-type medications, given the orange discoloration of the patient’s tissue sampling, similar to that of clofazimine use [60].

## 5. Conclusions

To our knowledge, we have presented the third case of non-crystallized IgSH. IgSH presents in diverse forms and locations, making the histological examination crucial in the process of identifying cases of this entity. It is likely underreported, given the diversity of the presentations and symptomatology. Chantranuwat et al. suggested in 2007 to use the term ISH widely, with a subclassification of crystallized or non-crystallized material [4]. We hereby propose to use the term IgSH to distinguish this form of Storing Histiocytosis from the abbreviation ISH, which in pathology is used for In Situ Hybridization. By our present literature review, we highlight the importance of investigating the presence of an underlying plasma cell dyscrasia or other B-cell malignancy when deposits of uncertain etiology are identified.

## Figures and Tables

**Figure 1 jcm-10-01834-f001:**
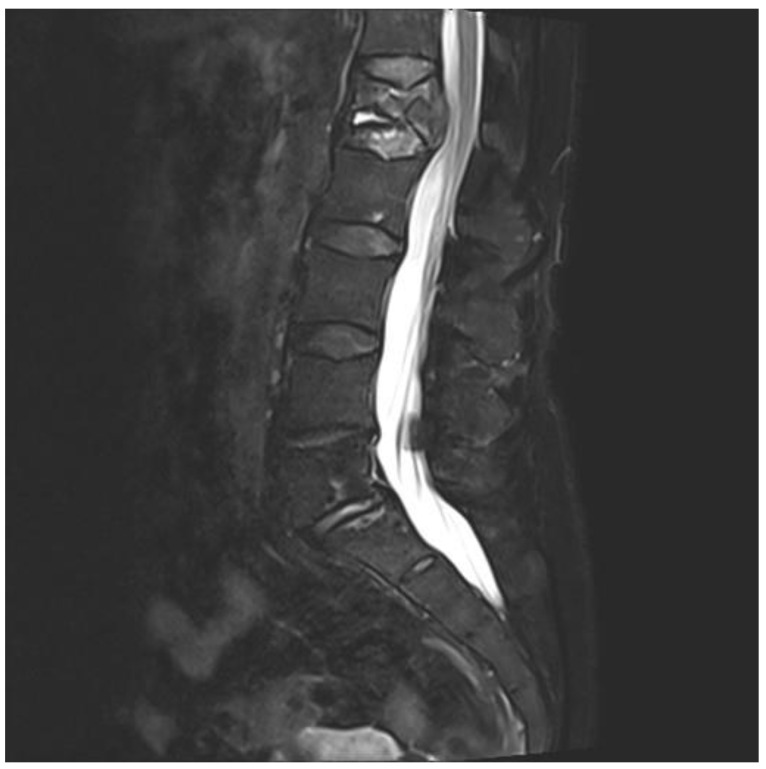
Magnetic resonance imaging (MRI) of lumbar column. The figure demonstrating collapse and fracture of vertebra corpus L1, with edema and compression against the spinal cord.

**Figure 2 jcm-10-01834-f002:**
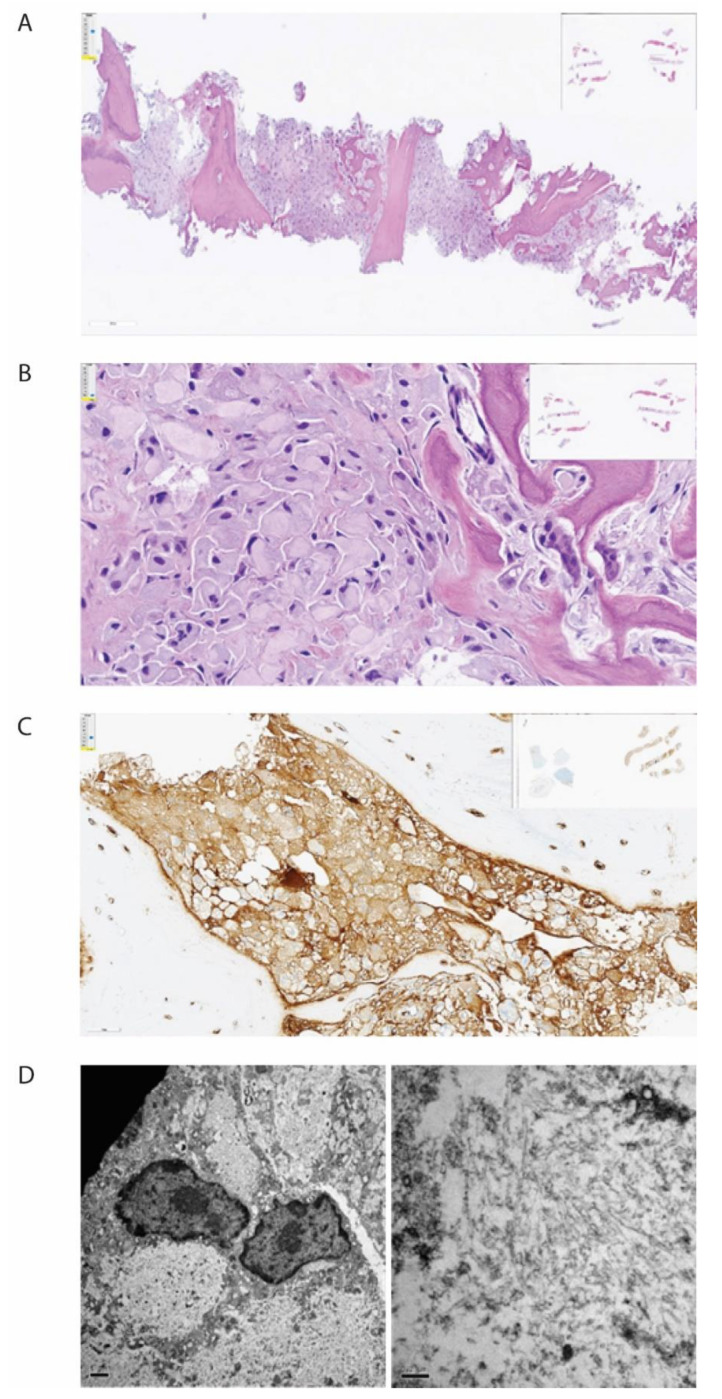
Histopathological examination of biopsy from corpus vertebrae. The figure demonstrates the biopsy from the patient’s corpus vertebrae L1. (**A**) Hematoxylin-eosin stain in low magnification demonstrating bone repair and bone marrow with the absence of organized hematopoiesis and maximal infiltration by histiocytes. (**B**) Hematoxylin-eosin stain in higher magnification demonstrating histiocytes with intracellular amorphous material. (**C**) Immunohistochemical staining for kappa light chain, demonstrating positivity in the histiocytes (brown color). (**D**) Electron microscopy, low magnification, showing bright intracellular material (**left**) and high magnification, showing no fibrils but spread tiny tubules (**right**).

**Figure 3 jcm-10-01834-f003:**
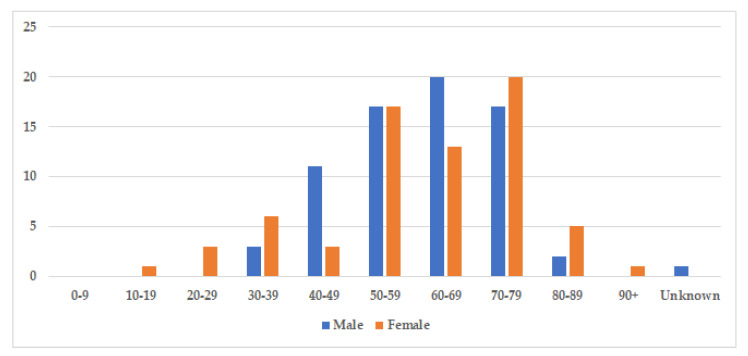
Age and gender distribution. The figure displays the incidence of storing histiocytosis in different decades. The median age at diagnosis was 60.5 years for both sexes (range 18–91, and slightly more cases were identified amongst men with 71 men (51%), compared to 69 women (49%).

**Figure 4 jcm-10-01834-f004:**
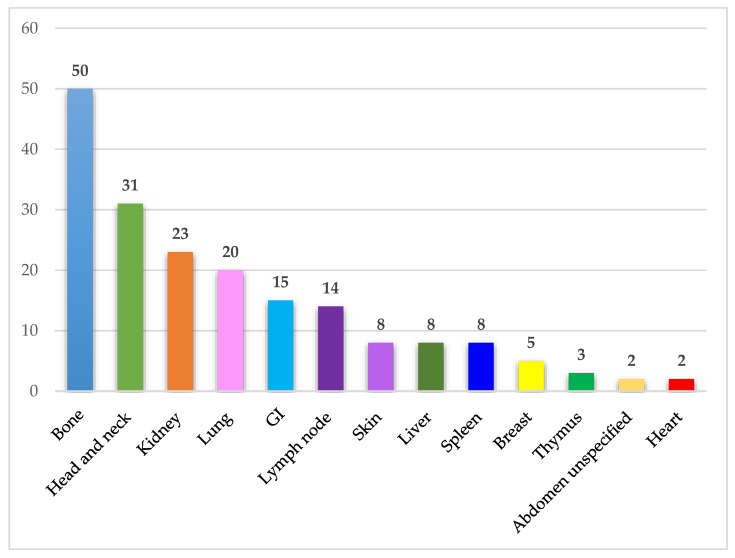
Affected organ systems. The figure demonstrates the number of identified cases in different organ systems. Abdomen unspecified includes one case of ascites and one case of a retroperitoneal mass. Abbreviations: GI—Gastrointestinal tract.

**Figure 5 jcm-10-01834-f005:**
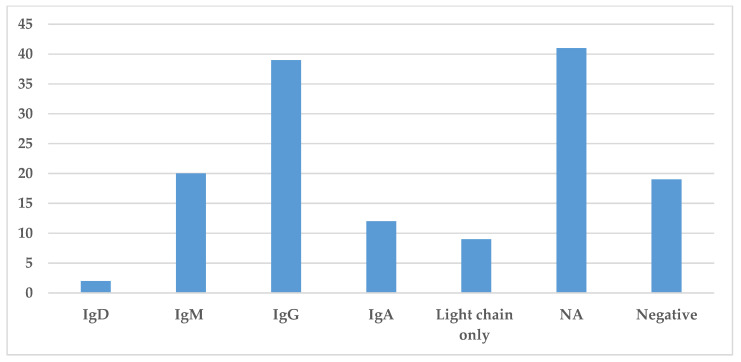
Prevalence of different immunoglobulins in Storing Histiocytosis. This figure demonstrates the distribution of immunoglobulin found in the serum by protein electrophoresis or immunofixation. Abbreviation: NA—not available.

**Figure 6 jcm-10-01834-f006:**
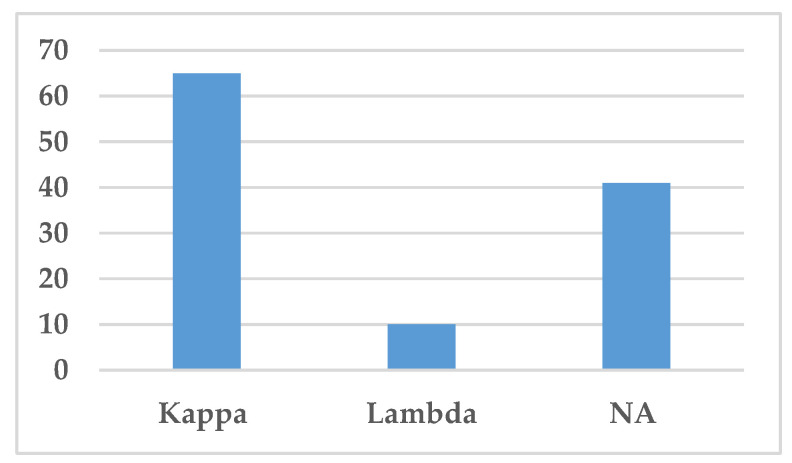
Distribution of light chains. This figure demonstrates the distribution of light chains found in the serum by protein electrophoresis or immunofixation. Abbreviation: NA—Not available.

**Figure 7 jcm-10-01834-f007:**
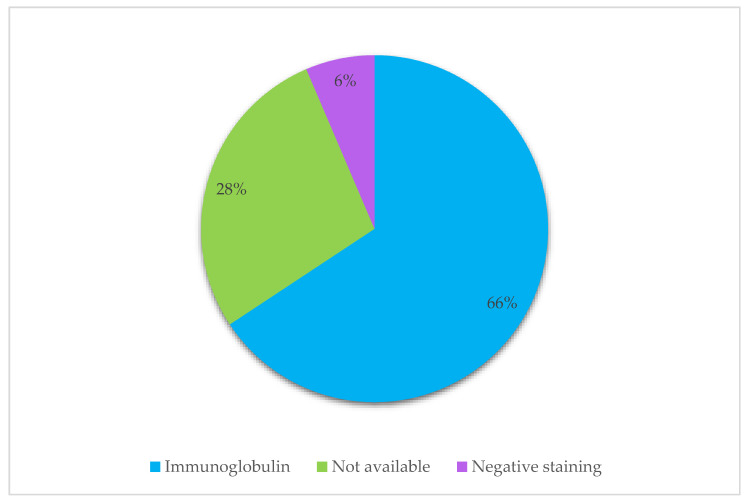
Distribution of the deposited material within histiocytes. The figure presents the distribution of material within histiocytes in identified cases, as described by immunohistochemical stains, to determine the origin of the deposited material.

**Table 1 jcm-10-01834-t001:** Diagnostic blood test from the patients.

Analysis	Values	References
Hemoglobin (g/dL)	10.1	11.7–15.3
EVF	0.33	0.35–0.46
CRP (mg/L)	18	<5
SR (mm/t)	69	1–30
Creatinine (µmol/t)	78	45–90
Protein (g/L)	69	62–78
IgG (g/L)	16.1	6.0–15.3
IgA (g/L)	0.76	0.8–4.0
IgM (g/L)	0.51	0.3–2.30
Kappa free light chains (mg/L)	54.0	6.7–22.4
Lambda free light chains (mg/L)	25.0	8.3–27.0
Ratio kappa/lambda free light chains	2.16	0.31–1.56
S-protein electrophoresis	Monoclonal band	
S-immunofixation	Monoclonal band type IgG kappa.	
M-protein (mg/L)	7.1	0

The table demonstrates different analyses performed, the obtained value, and the given reference area. Abbreviations: EVF, erythrocyte volume fraction; CRP, C-reactive protein; SR, sedimentation rate; Ig, immunoglobulin.

**Table 2 jcm-10-01834-t002:** Overview of literature search.

Case nr.	Organs/Tissue Sites													Age	Sex	Type	LP–PCD	Material within Histiocytes (ISH)	Serum Immunofixation	Ref.
	Skin	Bonemarrow	Kidney	GI-Tract	Liver	Abdomen unsp.	Heart	Thymus	Breast	Lung	Spleen	Head an Neck	Lymph Node							
1														71	M	L	MGUS	NA	IgG kappa	[6]
2														57	M	G	LPL	IgM/IgG/IgD kappa/lambda	IgG kappa	[7]
3														56	F	L	MM	Kappa	IgG kappa	[8]
4														56	M	L	MALT lymphoma	Kappa	NA	[9]
5														75	F	G	MM	Kappa	IgG kappa	[10]
6														66	F	L	MM	Kappa	IgA kappa	[11]
7														86	M	L	MZL	Negative	IgM	
8														68	F	L	NA	Kappa	Kappa	[12]
9														49	M	G	MM	NA	NA	
10														74	F	G	DLBCL	IgM kappa	IgM kappa	[13]
11														83	F	L	MM	Kappa	NA	[14]
12														73	F	G	MGUS	NA	IgG kappa	[15]
13														79	M	L	MM	Negative	NA	[16]
14														71	M	L	LPL	IgM Kappa	IgM kappa	[17]
15														67	M	L	MGUS	IgG kappa	IgG kappa	
16														74	M	L	MM	IgG kappa	IgG kappa	
17														65	M	G	None	Negative	IgG kappa	[18]
18														72	F	L	MM	Kappa	Kappa	[19]
19														36	M	L	None	Kappa	Negative	[1]
20														63	M	L	B-cell lymphoma	NA	IgM Kappa	[2]
21														40	M	L	MM	Kappa	Kappa	
22														71	M	G	MM	Kappa	IgG kappa	
23														65	F	L	MM	Kappa	IgA Kappa	
24														70	M	G	MM	NA	IgG Kappa	
25														73	M	L	MGUS	NA	IgG Kappa	
26														56	M	L	MM	NA	IgG Kappa	
27														60	M	G	MM	Kappa	IgG kappa	[20]
28														48	M	L	MM	Kappa	Kappa	[21]
29														70	M	L	MM	NA	IgG kappa	[22]
30														36	F	L	MALT lymphoma	Kappa	IgG	[23]
31														27	F	L	EMZL	Lambda	Lambda	[24]
32														67	M	L	MGRS	Kappa	IgG kappa	[25]
33														51	M	G	MM	IgG/lambda/kappa	IgG lambda	[26]
34														75	M	L	MM	IgG kappa	IgG kappa	
35														46	F	L	MM	NA	IgA kappa	
36														74	M	L	LPL	IgM	IgM	
37														63	M	L	LPL	Lambda	IgM lambda	
38														79	M	L	LPL	Kappa	IgM kappa	
39														43	M	G	EMZL	IgA kappa	Negative	
40														63	F	L	EMZL	IgM lambda	NA	
41														50	F	L	EMZL	IgM lambda	NA	
42														33	F	L	EMZL	Kappa	NA	
43														73	F	L	EMZL	NA	NA	
44														58	M	L	EMZL	NA	NA	
45														68	F	L	SMZL	Lambda	Negative	
46														56	M	L	None	NA	Negative	[27]
47														58	M	L	EMZL	Kappa	NA	[28]
48														46	M	L	MM	Lambda	NA	[29]
49														62	F	L	EMZL	IgM	NA	[30]
50														80	F	L	LPL	NA	IgM kappa	[31]
51														52	M	L	MM	Negative	IgG kappa	
52														57	F	L	MM	Kappa	IgA kappa	[32]
53														53	F	G	LPL	NA	IgM lambda	[33]
54														69	M	G	MGUS	NA	Kappa	[34]
55														NA	M	L	None	NA	NA	[35]
56														71	F	L	MALT lymphoma	IgG kappa	Negative	[36]
57														43	F	L	MGUS	Neg	IgD kappa	[37]
58														91	F	G	EMZL	IgM/IgG/kappa	IgM kappa	[38]
59														77	M	L	MZL	Kappa	Negative	[39]
60														53	M	L	MALT lymphoma	NA	NA	[40]
61														38	F	L	MM	IgA kappa	IgA kappa	[41]
62														30	F	L	None	NA	Negative	[42]
63														54	F	L	MALT lymphoma	Kappa	NA	[43]
64														72	F	L	BPDCN	NA	Negative	[44]
65														32	M	L	EMZL	NA	NA	[45]
66														55	F	L	None	IgA/IgG/IgM/kappa/lambda	Negative	[46]
67														54	F	L	MGUS	Kappa	NA	[47,48]
68														89	F	L	MZL	Kappa	NA	
69														50	F	L	MZL	Kappa	NA	
70														63	M	L	MM	Kappa	NA	
71														68	M	L	MGUS	IgG kappa	IgG kappa	
72														78	F	L	MGUS	NA	NA	[49]
73														80	M	L	None	NA	Negative	[50]
74														20	F	L	None	Lambda	NA	[51]
75														62	F	G	MGUS	NA	IgG kappa	[52]
76														64	M	L	EMZL	NA	NA	[53]
77														57	F	L	EMZL	NA	NA	
78														48	F	G	MGUS	Kappa	IgG kappa	[54]
79														51	F	L	MGUS	IgM/IgG	NA	[55]
80														38	M	L	None	IgG/kappa/lambda	Negative	[56]
81														32	F	L	None	Kappa	NA	[57]
82														50	F	L	LPL	NA	IgM kappa	[58]
83														67	M	L	MM	IgG	IgG	[59]
84														27	F	L	None	NA	NA	[60]
85														54	F	L	EMZL	Kappa	NA	[61]
86														75	F	L	MGUS	NA	IgG kappa	[62]
87														63	M	L	MM	NA	IgG kappa	[63]
88														53	F	L	MALT lymphoma	Non-crystallized IgG kappa	IgA kappa	[5]
89														52	M	L	MGUS	NA	IgM kappa	[64]
90														70	M	L	DLBCL	NA	IgM kappa	
91														65	M	L	MGUS	NA	IgG kappa	
92														64	M	L	MG	Negative	IgG kappa	[65]
93														76	F	L	MZL	IgM lambda	IgM lambda	[66]
94														66	M	G	MM	NA	IgG kappa	[67]
95														66	M	L	MGUS	Lambda	IgG lambda	[68]
96														65	M	G	None	NA	Kappa	[69]
97														54	M	L	MM	NA	IgG kappa	[70]
98														81	F	L	MALT lymphoma	IgM kappa	Negative	[71]
99														56	F	L	None	Kappa/lambda	Negative	[72]
100														52	M	L	MM	Non-crystallized kappa	NA	[4]
101														41	M	L	MM	NA	IgD kappa	[73]
102														62	F	L	MM	Kappa	IgG kappa	[74]
103														70	F	L	None	Kappa	Kappa	[75]
104														79	F	L	MM	IgA kappa	IgA kappa	[76]
105														69	F	L	MALT lymphoma	NA	Negative	[77]
106														49	M	G	MGUS	Kappa	IgG kappa	[78]
107														72	F	G	MM	Kappa	IgA kappa	
108														62	F	L	MM	NA	IgG kappa	[79]
109														74	F	L	MM	Negative	IgA lambda	[80]
110														51	M	G	MM	IgG kappa	Kappa	[81]
111														59	M	L	EMZL	IgG/IgM/kappa/lambda	NA	[82]
112														58	M	L	MZL	Lambda	IgM lambda	[83]
113														73	M	G	MM	IgA kappa	IgA kappa	[3]
114														62	F	L	EMZL	Kappa	Negative	[84]
115														48	M	L	MM	IgA kappa	IgA kappa	[85]
116														53	M	L	MM	IgA kappa	Negative	
117														50	M	G	MM	IgG kappa	Negative	
118														77	F	L	MM	Kappa	Negative	
119														66	M	L	MGUS	IgA kappa	IgA kappa	
120														66	M	G	LPL	IgM kappa	Negative	
121														68	F	L	LPL	IgM kappa	IgM/IgG/kappa	
122														53	F	L	LPL	IgM kappa	IgM kappa	
123														70	M	G	LPL	IgM kappa	NA	
124														70	M	G	LPL	IgM lambda	NA	
125														35	F	L	LPL	IgA kappa	IgG lambda	
126														54	F	L	None	IgG lambda	NA	
127														72	F	L	LPL	IgM kappa	NA	[86]
128														44	M	G	LPL	IgG kappa	IgG kappa	[87]
129														73	F	L	None	IgG/kappa/lambda	IgG	[88]
130														81	F	L	MALT lymphoma	IgM lambda	NA	[89]
131														61	F	L	LPL	IgM/IgG/kappa/lambda	IgM kappa	[90]
132														54	F	L	None	IgA/IgM/IgG/kappa/lambda	NA	[91]
133														46	M	L	LPL	Negative	IgG/IgM/lambda	[92]
134														49	M	L	B-cell lymphoma	NA	NA	[93]
135														78	F	G	LPL	Negative	IgM kappa	[94]
136														77	F	G	LPL	IgM kappa	NA	
137														18	F	L	LPL	Negative	NA	
138														75	M	G	MM	IgG kappa	NA	[95]
139														60	M	G	MM	IgA kappa	NA	[96]
140														75	F	L	MGUS	Non-crystallized kappa	IgG kappa	Pres

The table demonstrates the cases of storing histiocytosis we identified by literature review. The cases are numbered, and the different organ systems affected are identified by different colors in the left columns. To the right, we present patient age, sex, associated disorders, material identified, and results of immunofixation. Abbreviations: LP–PCD—Lymphoproliferative–Plasma cell disorder; M—Male; F—Female; L—Localized; G—Generalized; ISH—In-situ hybridization; MGUS—Monoclonal gammopathy of undetermined significance; MGRS—Monoclonal gammopathy of renal significance; MM—Multiple myeloma; LPL—Lymphoplasmacytic lymphoma; DLBCL—Diffuse large B-cell lymphoma; BPDCN—Blastic plasmacytoid dendritic cell neoplasia; MALT lymphoma—Mucosa-associated lymphoid tissue lymphoma; EMZL—Extranodal marginal zone lymphoma; SMZL—Splenic marginal zone lymphoma; MZL—Marginal zone lymphoma; NA—Not available.

**Table 3 jcm-10-01834-t003:** Demographic and pathological data from identified cases.

	Number (*n*)	Percentage (%)
Sex		
Men	71	51
Women	69	49
Etiology		
Localized	108	77
Generalized	32	23
1. SH with underlying LP–PCD	122	87
MGUS/MGRS	21	15
MM	43	31
LPL	21	15
DLBCL	2	1
BPDCN	1	1
MALT lymphoma	10	7
EMZL/SMZL	23	16
B-cell lymphoma not specified	2	1
2. SH without underlying LP–PCD	17	12
3. SH with unknown history	1	1

The table demonstrates the distribution of cases of storing histiocytosis regarding age, gender, localized versus generalized disease and the presence or not of an underlying lymphoproliferative or plasma cell disorder. Abbreviations: LP–PCD—lymphoproliferative–Plasma cell disorder; MGUS—Monoclonal gammopathy of undetermined significance; MGRS—Monoclonal gammopathy of renal significance; MM—Multiple myeloma; LPL—Lymphoplasmacytic lymphoma; DLBCL—Diffuse Large B-cell lymphoma; BPDCN—Blastic plasmacytoid dendritic cell neoplasia; MALT lymphoma—Mucosa-associated lymphoid tissue lymphoma; EMZL—Extra nodal marginal zone lymphoma; SMZL—Splenic marginal zone lymphoma.

**Table 4 jcm-10-01834-t004:** CSH classification according to crystal and etiology.

According to Crystal	According to Etiology
Immunoglobulin	Associated with LP–PCD
2. Charcot–Leyden	- MGUS
3. Clofazimine crystals	- MM
4. Others	- LPL
Cystinosis	Autoimmune
Silica-associated	- Sjogren’s disease
	- Rheumatoid arthritis
	Drugs
	- Clofazimine
	- Carbamazepine
	Reactive-inflammatory
	- Crohn’s disease
	Metabolic
	- Cystinosis

The table demonstrates classification systems according to crystal stored and etiology. Abbreviations: LP–PCD—lymphoproliferative–Plasma cell disorder; MGUS—Monoclonal gammopathy of undetermined significance; MM—Multiple myeloma; LPL—Lymphoplasmacytic lymphoma.

**Table 5 jcm-10-01834-t005:** Proposal of new classification.

New Classification
Immunoglobulin Storing Histiocytosis
- Crystallized
- Non-crystallized
Non-immunoglobulin Crystallized Storing Histiocytosis
- Charcot–Leyden
- Clofazimine
- Silica-associated
- Cystinosis

The table demonstrates a new proposed classification system according to immunoglobulin relation.

## Data Availability

The data presented in this study are available in Table 2 in the present article.

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
