# Peer review of "Immunoglobulin-Storing Histiocytosis: A Case Based Systemic Review"

_jcm, 2021, doi:10.3390/jcm10091834_

Round 1

Reviewer 1 Report

The authors report immunoglobulin-storing histiocytosis.

  1. In this study, there is no information of treatment strategy and clinical outcome. The authors should describe these things in the text.

Author Response

We are grateful for this comment regarding treatment and prognosis, and accordingly we have added a section in the discussion part of the manuscript regarding treatment and prognosis.

Reviewer 2 Report

Thank you for letting me reviewing this article

1/ cas report : the clinical case report is well described and documented.

  • how did you conclude that IgSH was localized ? Did you perform MRI or 18-FDG PET CT? How do you explain anemia?
  • Did you discuss a systemic treatment?
  • Can you precise if the amorphous material was light chain  or complete immunoglobulin?

2/ litterature review

This an extensive and very intersting litterature review.

  • would it be possible to have more information about systemic treatment and  its efficiency?
  • some patients have systemic symtomes revealing CSH as fever or inflammatoy syndrom. May be it should be included in the description

3/ discussion

Discussion is very long. I think it is not necessary to detail the clinician attitude in front of  back pain. However it would be intersting to discuss how your new classification will influence the treatment for the patients ? Do crystalized and non crystallized CSH have different evolution?

Author Response

Thank you for letting me reviewing this article

1/ case report: the clinical case report is well described and documented.

  • how did you conclude that IgSH was localized ? Did you perform MRI or 18-FDG PET CT? How do you explain anemia?
  • Did you discuss a systemic treatment?
  • Can you precise if the amorphous material was light chain  or complete immunoglobulin?

MRI scan was performed, although PET CT was not performed. We have increased the discussion of the features of MRI findings briefly in our revised version. Whole-body low-dose computed tomography (CT) for assessment of osteolytic lesion as part of multiple myeloma was performed, without detecting other lesions. And since no other signs of disease was detected, we conclude this was localized disease. Hemoglobin was never <10 g/dl, and hence we conclude with no obvious signs for anemia.

Systemic treatment was discussed, although the patients did not fulfill the diagnostic features for multiple myeloma (MM), we therefore considered that the patient would have no sure benefit of systemic treatment.

The amorphous material was considered to be crystallized light chain. We have discussed these features in more dept in our revised version.

2/ literature review

This an extensive and very interesting literature review.

  • would it be possible to have more information about systemic treatment and its efficiency?
  • some patients have systemic symptoms revealing CSH as fever or inflammatory syndrome. Maybe it should be included in the description

Regarding, systemic treatment, this is also requested by reviewer 1, and accordingly we included more information about systemic treatment.

We are also grateful for the comments regarding systemic symptoms and a general inflammatory condition seen in some of these patients, and accordingly we have added this in our discussion part as well.

3/ discussion

Discussion is very long. I think it is not necessary to detail the clinician attitude in front of back pain. However it would be interesting to discuss how your new classification will influence the treatment for the patients? Do crystalized and non-crystallized CSH have different evolution?

We agree in these comments, and accordingly we have shortened the general discussion part regrading general diagnostic work up of lower back pain etc., and instead we have added some details about systemic treatment and classification system for CSH, and potential impact of treatment decisions.

Reviewer 3 Report

The authors have presented a case of immunoglobulin-storing histiocytosis, and its association with monoclonal gammopathy, especially with kappa light chain expression, clinical findings, diagnostic workup, and therapeutic options, through a systematic review of the literature regarding IgSH.

The paper is written in a good English, and it can be clear for readers.

Into introduction/discussion, adding a table with results of the main papers in this topic could be really useful.

However, the idea is good and well presented and the paper could be really interesting for readers, and it can be continued after minor revision.

Author Response

Reviewer 3

The authors have presented a case of immunoglobulin-storing histiocytosis, and its association with monoclonal gammopathy, especially with kappa light chain expression, clinical findings, diagnostic workup, and therapeutic options, through a systematic review of the literature regarding IgSH.

The paper is written in a good English, and it can be clear for readers.

Into introduction/discussion, adding a table with results of the main papers in this topic could be really useful.

However, the idea is good and well presented and the paper could be really interesting for readers, and it can be continued after minor revision.

We are grateful for the mainly positive comments regarding our presented paper from reviewer 3. Given the rarity of the disease, which mostly are based on case reports/serial case reports, and the lack of epidemiological and clinical studies, it is difficult to present major papers. We believe we have discussed all the cases describe in the major Figure 3. Although, we have more clearly tried to summarize the major established knowledge in the field, with key references, in the first part of the discussion part of the article, to make it easier from the readers to follow the topics.

Round 2

Reviewer 2 Report

Thank for your answer. This is an intersting case report and review.